# Evaluating HPV Viral Load and Multiple Infections for Enhanced Cervical Cancer Risk-Based Assessment

**DOI:** 10.3390/life15020153

**Published:** 2025-01-22

**Authors:** Serena Varesano, Giulia Ciccarese, Michele Paudice, Katia Mazzocco, Gabriele Gaggero, Simone Ferrero, Giancarlo Icardi, Valerio Gaetano Vellone

**Affiliations:** 1Hygiene Unit, IRCCS Ospedale Policlinico San Martino, Largo R. Benzi 10, 16132 Genoa, Italy; icardi@unige.it; 2Section of Dermatology, Department of Medical and Surgical Sciences, University of Foggia, Viale Pinto 1, 71122 Foggia, Italy; giulia.ciccarese@unifg.it; 3Department of Integrated Surgical and Diagnostic Sciences (DISC), University of Genoa, Viale Benedetto XV 14, 16132 Genoa, Italy; michele.paudice@unige.it (M.P.); valerio.vellone@unige.it (V.G.V.); 4Pathology University Unit, IRCCS Ospedale Policlinico San Martino, Largo R. Benzi 10, 16132 Genoa, Italy; 5Pathology Unit, IRCCS Istituto Giannina Gaslini, Via G. Gaslini 5, 16147 Genoa, Italy; katiamazzocco@gaslini.org (K.M.); gabrielegaggero@gaslini.org (G.G.); 6Obstetrics and Gynecology University Unit, IRCCS Ospedale Policlinico San Martino, Largo R. Benzi 10, 16132 Genoa, Italy; simone.ferrero@me.com; 7Department of Neurosciences, Rehabilitation, Ophthalmology, Genetics, Maternal and Child Health (DINOGMI), University of Genoa, Largo Paolo Daneo 3, 16132 Genoa, Italy; 8Department of Health Sciences (DISSAL), University of Genoa, Via A. Pastore 1, 16132 Genoa, Italy

**Keywords:** cervical cancer, human papillomavirus, HPV screening, second-level cervical cancer screening, viral load, co-infection, cervical intraepithelial neoplasia, HPV from formalin-fixed paraffin-embedded blocks (FFPE), Italy

## Abstract

Human papillomavirus (HPV) is a major cause of cervical cancer, a significant health concern worldwide. Despite advances in screening methods, including the Pap test and the HPV DNA test, limitations remain in accurately predicting which HPV infections will progress to cervical intraepithelial neoplasia (CIN) and, eventually, invasive cancer. This study evaluates the usefulness in real life of assessing HPV viral load and the presence of multiple HPV genotypes in enhancing the diagnostic accuracy of triage in cervical cancer screening. A retrospective analysis was performed on 55 formalin-fixed paraffin-embedded cervical samples collected from women who underwent colposcopy with a biopsy or conization at San Martino Hospital, Genova, Italy, between January and June 2021. Histological diagnoses were compared with molecular analyses (HPV genotyping, viral load quantification and co-infection) using a multiplex real-time PCR platform. Of the samples analyzed, 56.4% were HPV DNA positive, while 40% tested negative. The molecular analysis identified more HPV-negative cases than the histological analysis (*p* < 0.05). Higher viral loads and HPV co-infections were more frequent in high-grade CIN lesions. These markers may help identify patients at an elevated risk for persistent infections and cancer progression. These findings support the potential of integrating HPV viral load and genotype co-infection assessments into routine cervical cancer screening protocols to improve early detection and reduce overtreatment and unnecessary interventions.

## 1. Introduction

Human papillomavirus (HPV) is the most common sexually transmitted infection (STI) worldwide, affecting most sexually active individuals at least once in their lifetime [1,2,3]. Over 200 HPV genotypes have been identified, with 40 classified as oncogenic, primarily linked to cancers of the lower genital tract, including cervical, anal, and oropharyngeal cancers [1]. High-risk (HR) HPV types, particularly HPV16 and HPV18, account for over 70% of cervical cancers and can progress from precancerous lesions to invasive cancer if not cleared by the immune system [1,2]. The virus’s progression includes integration into host DNA, where HR-HPV oncogenes E6 and E7 promote cellular DNA damage, leading to dysplasia and cancer. DNA integration causes the overexpression of p16, a cyclin-dependent kinase inhibitor, in dysplastic cervical cells (Figure 1). This overexpression serves as a biomarker of viral genome integration and is readily detectable in cervical tissue through immunohistochemistry [4].

Cervical intraepithelial neoplasia (CIN) represents precancerous changes in cervical cells. CIN is classified as mild (CIN1), moderate (CIN2), or severe (CIN3), reflecting increasing risks of progression to cancer. While most CIN1, a low-grade squamous intraepithelial lesion (LSIL), regress without treatment, CIN2 and CIN3 are high-grade lesions (HSILs) and carry a significant progression risk of up to 30% over 30 years without treatment [5]. In this study, cervical intraepithelial neoplasia (CIN) was classified as an LSIL (CIN 1) or HSIL (CIN 2/3) according to standard terminology (2001 and 2014 Bethesda System) [6,7].

Certain risk factors increase the likelihood of cancer in women with high-risk HPV infections. These include immunosuppression (due to HIV or certain drugs), recipients of solid organ transplants, smoking or secondhand smoke, long-term use of oral contraceptives, and obesity (BMI ≥ 30 kg/m^2^) [8].

Effective screening programs are critical for detecting precancerous lesions early, especially in individuals with known risk factors. Cervical cancer is preventable by HPV vaccination [9], routine cervical cancer screening, and appropriate follow-up and treatment when needed [10]. The World Health Organization (WHO) recommends screening all women over 25–30 years old with a Pap test and an HPV DNA test to identify precancerous lesions, which are usually asymptomatic, before they progress to invasive cancer [4].

While essential, Pap and HPV DNA tests have notable limitations. The Pap test, with its variable specificity, often produces false-positive results, leading to unnecessary follow-up testing and increased anxiety for patients. Similarly, the HPV DNA test, although effective at detecting high-risk HPV genotypes, cannot reliably differentiate between transient infections and those likely to progress to cancer, as the majority of HPV infections are non-malignant and resolve spontaneously [11,12].

In Italy, cervical cancer screening is organized at the regional level, with each region autonomously defining its guidelines and protocols. The program operates on two levels: primary and secondary level screening. National and international recommendations are adapted locally based on regional resources, demographic factors, and territorial characteristics, ensuring protocols are evidence-based and tailored to the specific needs of each region [13].

First-level cervical cancer screening in Italy typically involves the Pap test or the HPV DNA test, with the latter recommended as the primary strategy [13,14]. Its adoption varies across regions due to differences in healthcare organization and resources, with some regions fully implementing it and others transitioning from older protocols. Women with abnormal results are referred to triage procedures, such as colposcopy and biopsy, for detailed evaluation and treatment planning, though these methods are not entirely foolproof.

Non-neoplastic anomalies, such as cervical atrophy, metaplasia, or inflammation, can resemble LSILs, leading to overdiagnosis or unnecessary treatment. In this context, there is a growing need for more accurate and individualized diagnostic approaches [13,14].

Although HPV infection is a necessary factor, it is insufficient for the malignant transformation of the cervix. Similarly, cytology and histology alone cannot determine the progression risk of low-grade lesions [15,16]. HR-HPV DNA testing at recommended intervals of 3–5 years, along with regular Pap testing, is effective for the early detection of preneoplastic lesions [13,14]. However, to enhance the diagnostic accuracy of *second-level* cervical cancer screening, there is increasing interest in integrating molecular markers such as HPV viral load and co-infection status. The assessment of HPV viral load could offer insights into the persistence of infection, as higher viral loads are often associated with sustained infections that pose a greater risk of progression. Additionally, the presence of multiple HPV genotypes (co-infections) may influence lesion development, with some studies suggesting that co-infections may increase the likelihood of progression to high-grade lesions [17,18].

Emerging diagnostic approaches like mRNA analysis, E4 protein detection, dual staining (DS), and methylation tests show promise for improving diagnostic accuracy [19,20]. Among these, mRNA analysis is particularly valuable for identifying biologically active HPV infections. By detecting E6/E7 mRNA, it provides critical insights into oncogenic activity following viral genome integration, helping to identify high-risk infections likely to progress. Despite its clinical potential, mRNA analysis remains underutilized in practice [21,22,23].

Therefore, in addition to the morphological and molecular analyses, other molecular and epigenetic factors such as the simultaneous infections by multiple HR HPVs, the HPV load [4,24] and other concomitant STIs [25,26,27], may be assessed to evaluate the risk of the persistence of the infection and the cancer progression.

This study aims to enhance the diagnostic accuracy of triage cervical cancer screening by integrating molecular analyses, such as HPV viral load and genotype co-infection, with conventional morphological examinations. This combined approach seeks to improve the identification of high-risk lesions, distinguish true infections from non-neoplastic abnormalities, and more accurately predict the risk of lesion persistence or progression. By incorporating molecular diagnostics, this study aims to refine patient stratification, minimize colposcopy clinics’ overload and unnecessary follow-up procedures, and enable more targeted and efficient clinical interventions for cervical cancer prevention.

The integration of HPV viral load and genotype co-infection assessments holds significant promise for improving diagnostic accuracy in routine *second-level* cervical cancer screenings. This approach has the potential to refine risk stratification and facilitate more personalized clinical interventions in real-world settings.

## 2. Materials and Methods

### 2.1. Tissue Specimens

This retrospective study evaluated histological samples from patients who underwent colposcopy with a biopsy or conization at the San Martino Hospital, Genoa, Italy, between 1 January 2021 and 3 June 2021. The timeframe chosen was selected to ensure a manageable and consistent sample cohort representative of our clinical setting.

To replicate the “*real-life*” utility of molecular analysis in routine clinical practice, we applied specific inclusion criteria to selectively sample patients undergoing triage after abnormal Pap or HPV DNA test results. This approach was designed to align with routine diagnostic workflows and ensure the applicability of our findings to everyday clinical practice.

Selection criteria included patients being at the *second-level* screening stage with abnormal Pap or HPV DNA test results and adequate biopsy material to perform simultaneous molecular and histological analyses. No patients with known concurrent malignancies were included in this study, ensuring the focus remained solely on cervical pathology. Additionally, clinical information regarding the smoking or alcohol consumption histories of the selected patients was not available. Only cases with complete and high-quality histological and molecular data were included to maintain data reliability (Figure 2).

All patients received a histological diagnosis by an examination of hematoxylin and eosin-stained sections combined with the immunohistochemistry (IHC) of p16INK4a and Ki-67, as required by the routine analysis protocol, at the Histopathology and Cytopathology Unit.

We therefore studied the formalin-fixed paraffin-embedded (FFPE) cervical specimens of various sample types (cervical biopsies and conizations) diagnosed at the Division of Histopathology and Cytopathology of Hospital Policlinico San Martino in Genoa, Italy. The experimental molecular analysis of genotype, co-infections, and viral load was developed and performed at the Hygiene Unit. Each sample was analyzed for genotyping, determination of viral load, and co-infections through a multiplex real-time polymerase chain reaction (PCR) platform capable of simultaneously detecting, differentiating, and quantifying 28 HPV genotypes (19 HR-HPVs and 9 LR-HPVs).

### 2.2. Immunohistochemistry

Immunohistochemical staining was performed using the automated Ventana BenchMark XT immunostainer (Ventana Medical System Inc., Tucson, AZ, USA) according to the established protocol. The expressions of p16INK4a and Ki-67 were analyzed with IHC using anti-p16INK4a (monoclonal, clone E6H4, Roche, Basel, Switzerland) pre-diluted antibody and anti-human Ki-67 (monoclonal, Rabbit clone anti-Human Ki-67 SP6, Roche) pre-diluted antibody, respectively.

For analysis, the dysplastic epithelium was divided into thirds (basal to luminal). p16INK4a positivity was defined as intense, with diffuse nuclear and cytoplasmic staining extending at least to the middle third (block positivity). Only nuclear staining was considered for Ki-67 [4].

### 2.3. Molecular Analyses

Four 3 μm-thick sections were cut from the FFPE tissue of all samples and placed in an Eppendorf Tube^®^ of 1.5 mL. Subsequently, deparaffinization was performed with the MagCore^®^ Genomic DNA FFPE One-Step Kit (RBC Bioscience Corp., New Taipei City, Taiwan) following the manufacturer’s instructions (cartridge code: 405; execution time: 16 h; elution volume: 60 μL) using the automatic extractor MagCore^®^ HF16 Plus nucleic acid (RBC Bioscience Corp.). Genotyping, viral load, and co-infections were detected by Anyplex II HPV28 kit (Seegene, Seoul, South Korea), which simultaneously identified 28 genotypes: 19 high-risk HPV types (16, 18, 26, 31, 33, 35, 39, 45, 51, 52, 53, 56, 58, 59, 66, 68, 69, 73, 82) and 9 low-risk HPV types (6, 11, 40, 42, 43, 44, 54, 61, 70) by performing a multiplex PCR with a CFX96 thermal cycler (Bio-Rad, Hercules, CA, USA) [28,29]

Results were interpreted with Seegene Viewer software (version 3.28.000), which provided a final report indicating HPV genotype(s), the presence or absence of infection, and semi-quantitative viral load (categorized as weak (+), medium (++), or high (+++)).

The semi-quantitative viral load evaluation is based on the number of fluorescence detections from melting curve analysis. During analysis, the thermal profile provides three opportunities to measure fluorescence at set melting temperatures, specifically at steps 8, 14, and 20.

For each HPV genotype tested, if fluorescence related to a genotype is detected at all three temperature points, the viral load is classified as high (+++); detection at two temperature points indicates a moderate viral load (++); detection at only one temperature point signifies a low viral load (+); and if no fluorescence is detected across the three points, the viral load is considered undetectable (–) or negative [29,30].

The data collection and analysis adhered to established protocols, with the systematic handling of missing or invalid data to enhance the study’s reliability and reproducibility.

### 2.4. Statistical Analysis

To minimize potential selection and information biases, participants were selected using standardized criteria, including prior abnormal Pap or HPV DNA results, and histological assessments were conducted with pre-defined diagnostic protocols to ensure consistency. Additionally, rigorous inclusion criteria and the systematic handling of missing or invalid data were applied to enhance the reliability and reproducibility of the findings.

The statistical analysis aimed to evaluate the association between histological findings, HPV viral load, and the presence of multiple HPV genotype infections in cervical samples. Descriptive statistics were first used to summarize the demographic characteristics of the 55 patients and the distribution of histological diagnoses, which included normal tissue, LSILs, HSILs, and condylomatosis. The frequency of HPV-positive and HPV-negative cases, as well as the prevalence of high-risk (HR) and low-risk (LR) HPV genotypes, were also documented.

Data were anonymized and entered into a Microsoft Excel™ spreadsheet, with statistical analysis performed using MedCalc™ software (version 23.1.3). Fisher’s exact test was employed due to the small sample size to examine relationships between categorical variables. This test assessed whether there were significant differences in the prevalence of HPV positivity across different histological categories, and whether multiple HPV infections and high viral loads were associated with grade lesions such as HSILs. A 95% significance level was chosen. Continuous numeric variables are presented as mean ± standard deviation, and discrete variables as the number of cases (percentage).

The study was performed in line with the Helsinki Declaration and our institutional regulations. Each participant provided written informed consent which was approved by the Liguria Regional Ethics Committee (P.R. 162REG2017).

This work complies with STROBE guidelines.

## 3. Results

A total of 55 FFPE cervical samples collected from 55 women aged between 26 and 74 years (median age: 40.5 years) were studied. These samples were collected through cervical biopsies (42) and conization (13) (Table 1).

The flow diagram depicts patient sample management from initial collection through processing and analysis and provides a clear, structured overview of sample handling and exclusions, ensuring transparency in alignment with STROBE guidelines (Table 2).

### 3.1. Histological vs. Molecular Analysis

**Histological analysis** revealed normal cervical tissues in 5/55 cases (9.1%) and abnormal cervical tissues in 50 (90.9%). Among the 50 samples of abnormal cervical tissues, 26 cases were classified as LSILs, 23 as HSILs and 1 as cervical condylomatosis (Table 1, Figure 3A). Among the 5 tissues that resulted histologically normal, 2 cases were HPV DNA positive, 2 cases were HPV DNA negative and 1 case was invalid (Figure 3B).

**Molecular analyses** revealed positivity for HPV DNA in 31 cervical samples (56.4%), whereas 22 specimens were HPV-negative (40%), and 2 samples (3.6%) were invalid due to the paucity of material (Table 1, Figure 4A). Therefore, molecular analysis detected a higher number of negative cervical samples (22/55) compared to histological analysis (5/55). The difference in the detection of negative/normal samples was statistically significant (*p* < 0.05). It is important to note that the 22 cervical specimens negative for HPV DNA were classified as abnormal based on histological diagnosis, which included at least LSILs in 15 cases and HSILs in 5 cases. Chronic cervicitis without concurrent SILs was not included in the classification of abnormal findings.

Only 2 cases histologically evaluated as normal tissue were confirmed by molecular analysis. (Figure 4B).

In this study, 3 out of the 55 collected cervical samples were excluded from analysis due to insufficient material, which prevented both histological and molecular assessments. This exclusion primarily involved samples where the tissue quantity was inadequate to reliably perform both diagnostic procedures.

Such exclusions could introduce selection bias, as the omitted cases may differ in some way from those analyzed, potentially affecting the generalizability of the findings. For instance, if these insufficient samples had specific characteristics (e.g., lower lesion severity or different patient demographics), the final analyzed group may not fully represent the broader population intended for screening. Recognizing these exclusions and their potential effects on the study outcomes enhances transparency and helps interpret the results with an awareness of possible bias sources.

### 3.2. HPV Genotype and Histological Diagnosis

Overall, in the 31 HPV-positive cervical samples, 21 different HPV genotypes have been detected: 15 HR HPV genotypes (16, 18, 31, 33, 39, 51, 52, 53, 56, 58, 66, 68, 69, 73, 82) and 6 LR HPV genotypes (6, 11, 40, 42, 54, 61) (Table 2, Figure 5A). Figure 5B describes the HPV genotypes found in the cervical samples, their frequency, and the histological diagnosis to which each genotype was associated.

HPV16 was the most frequently encountered genotype in our cohort and was commonly associated with moderate/high-grade lesions (HSILs). Among the 5 cases corresponding to normal cervical tissue (negative for dysplasia/neoplasia), 2 were associated with HPV infection (1 with HR and 1 with LR HPV), 2 were negative for HPV, and in 1 case there was not sufficient tissue for molecular analysis (Figure 3B).

The positivity of LR and/or HR HPV infections in the different histological samples has been described in Table 3 and Figure 5. More specifically, among the samples that were positive for only HR-HPV DNA (24 cases), 7 were histologically diagnosed as LSILs, 16 as HSILs, and 1 as normal cervical tissue. The unique cervical sample harboring only LR HPV infection was histologically diagnosed as normal tissue. The six samples that resulted positive for both LR and HR HPV infections were histologically diagnosed as LSILs (3 cases), HSILs (2 cases), and condylomatosis (1 case) (Table 3, Figure 6).

The rate of HSILs was higher among cases harboring only HR HPV genotypes compared to cases harboring both HR and LR HPV genotypes; however, this difference was not statistically significant (*p* > 0.05).

### 3.3. Co-Infections, Genotypes, and Viral Load

Table 4 describes the histological diagnoses associated with cases characterized by multiple HPV infections, the genotypes of the multiple concurrent HPV infections in the same sample and the viral load in a semi-quantitative way. Ten of the 55 samples (18%) simultaneously harbored multiple types of HPV, most of them with a high viral load (in 6 out of 10 cases the HR HPV load was quantified as ++). Of the samples analyzed, 60% (6 out of 10) contained both high-risk (HR) and low-risk (LR) HPV genotypes, while the remaining 40% (4 samples) had multiple HPV infections, all of which were high-risk (HR) genotypes (Table 4, Figure 7).

The rate of HSILs was not higher among the samples harboring multiple concomitant HPVs (5/10 samples) compared to those harboring only one HPV infection (13/21 samples); however, this difference was not statistically significant (*p* > 0.05).

Among the cases with multiple concomitant HPV infections (Table 4, Figure 7), only 1 of the 4 LSIL cases harbored an HR HPV infection with a high load (++); conversely, all the 5 HSIL cases harbored at least one HR HPV genotype with a high load (++).

Moreover, it was evident that the 2 cases of HSILs were the only samples in which both HR HPV infections had a high viral load (++) (cases 8 and 9). All the other samples harboring multiple HR HPVs had one HPV with a high load (++) and the other HPV with a low load (+).

Unfortunately, the small number of cases with multiple HPV infections in our series (10 cases) limits the ability to perform statistical analysis or draw meaningful conclusions. However, with the increasing adoption of extended HPV genotyping, more published data are expected to shed light on the role of multiple HPV genotypes in CIN development.

## 4. Discussion

Integrating HPV viral load assessment and multiple genotype analysis into triage protocols represents a significant advancement in clinical and public health practices. Identifying higher viral loads or multiple HPV infections enables more precise risk stratification for persistent infections and lesion progression. This differentiation optimizes patient management by prioritizing follow-up for high-risk cases while minimizing overtreatment in low-risk cases. Refining colposcopy and biopsy referral criteria through molecular markers ensures a more efficient healthcare resource allocation, particularly in systems where screening demand may strain capacity. Implementing these findings into routine practice would not only enhance diagnostic accuracy but could also positively impact public health by reducing the psychological and economic burdens associated with unnecessary procedures, thereby aligning with the goals of value-based healthcare. These preliminary results advocate for further prospective studies to validate the utility of HPV viral load and co-infection status as reliable biomarkers, supporting evidence-based adjustments to cervical cancer screening guidelines.

One of the key strengths of this study is its focus on a molecular approach to *second-level* cervical cancer screening, moving beyond the traditional reliance on cytological and histological assessments. This dual approach ensures a comprehensive analysis of cervical samples, addressing the limitations of conventional screening methods that may overlook the cases of HPV persistence or overestimate the severity of lesions, particularly LSILs, which often reflect non-neoplastic abnormalities rather than true precancerous changes. By incorporating molecular diagnostics alongside morphological analyses, this study provides a richer dataset for understanding the complexity of HPV-related cervical pathology.

Recognizing the challenges of obtaining material for PCR testing from paraffin blocks, we employed a 16 h deparaffinization protocol combined with an automated extraction method based on magnetic bead technology. This routine real-time PCR approach has proven efficient and reliable. In our study, the extraction method demonstrated remarkable quality, with only 2 out of 55 samples being invalid, underscoring the robustness of the system and protocol. Based on our experience, we confidently recommend this extraction method and protocol for studies or analyses requiring high-quality DNA from formalin-fixed paraffin-embedded (FFPE) tissue samples.

In our analysis, nearly half (40%) of the cervical samples collected from women undergoing *second-level* screening were negative at the molecular level. This highlights the limitations of relying solely on morphological methods, as non-neoplastic abnormalities such as metaplasia, inflammation, and atrophy can mimic LSILs, complicating diagnosis and increasing the risk of overdiagnosis [17]. HPV-negative lesions classified as LSILs or HSILs in our study may reflect either misclassification during histopathological evaluation, particularly in ambiguous cases, or lesions in regression. A combined morphological and molecular approach, as demonstrated here, significantly improves lesion characterization, reduces overdiagnosis, and better stratifies patients, particularly when viral integration is undetectable [17,18]. For example, among the 22 HPV DNA-negative samples, only 2 were also negative on histopathological examination (normal cervical tissue), while most (15/22) were classified as LSILs.

In a previous study, more than 1000 cases of samples that were previously classified as CIN were blindly reviewed by 2 independent pathologists [17]. One of the major sources of variability was the marked tendency of the reviewers to downgrade the original interpretations from LSIL to negative. Although LSILs are often overinterpreted in cervical pathology practice, in the study by Della Palma et al. most of the LSIL cases downgraded at the pathologist review were HPV DNA positive, reflecting problems in establishing specific criteria for recognizing the morphologic features of the HPV cytopathic effect [17].

Similarly, in our smaller case series, among the 5 histologically normal tissues, 2 cases were HPV DNA positive, while the remaining 3 included 2 HPV DNA-negative samples and 1 invalid result. Therefore, our findings reaffirm the challenges in defining thresholds between normal tissue and LSILs [17,18].

The findings also provide valuable insights into multiple HPV genotype infections and their potential cumulative effects on cervical lesion progression. Although our series showed a slightly lower rate of multiple HPV genotypes in CIN (18%) compared to the literature reports (20–40%), the presence of high viral loads in cases of HSILs and their association with infection persistence underscore the diagnostic value of these molecular markers [31]. Concerning the viral load, in our series, it was calculated in a semi-quantitative way on the 10 samples characterized by multiple co-infections: a high viral load was detected in all the cases of HSILs and only one of the 4 LSIL cases (Table 3). Our data are in line with a Mexican study that found an association between multiple HPV genotypes’ cervical infections and high viral loads with infection persistence and, therefore, possible cancer development [24].

Furthermore, HPV16 was the most frequently encountered genotype in our cohort, commonly associated with moderate- and high-grade lesions (HSILs), confirming previous studies [1,2]. The declining prevalence of HPV18, detected in only 3 of 55 samples and all associated with HSILs, aligns with recent research and may reflect the herd effects of HPV vaccination. These findings emphasize the ongoing importance of HPV18 in advanced cervical pathology despite its reduced prevalence [32,33].

Lastly, a comment regarding the comparison between the Seegene Anyplex system and Roche Cobas. While the Cobas system offers a well-established correlation between viral load and cycle threshold (Ct) values, the Seegene Anyplex system stands out for its ability to provide extended HPV genotyping and simultaneously report viral loads for multiple genotypes. This feature enables a deeper understanding of co-infections and their role in lesion progression, enhancing diagnostic accuracy. Therefore, the Anyplex system is particularly useful for advanced screening and research settings, complementing the established capabilities of the Cobas system [34,35,36,37,38].

The present study offers several strengths and makes a significant contribution to the field of cervical cancer screening by focusing on the role of HPV viral load and multiple HPV genotype infections as diagnostic markers. One of the primary strengths of this study is its emphasis on a molecular approach to *second-level* cervical cancer screening, which goes beyond the traditional reliance on cytological and histological assessments. By integrating molecular diagnostics, this study addresses the limitations of conventional screening methods that may miss cases of HPV persistence or overestimate the severity of lesions, particularly LSILs, which can often reflect non-neoplastic abnormalities rather than true precancerous changes.

Furthermore, while this study offers significant contributions, several limitations must be acknowledged.

## 5. Study Limitations

The relatively small sample size (55 cases) limits the statistical power and generalizability of our findings, affecting the ability to draw robust conclusions and adequately represent the wider population. We acknowledge this limitation and emphasize that this is a preliminary study, intended to pave the way for larger, multicenter research to validate these findings and provide a more comprehensive understanding of the studied parameters. Larger cohorts are necessary to confirm these observations and refine screening protocols based on molecular markers.

Additionally, the retrospective nature of our study introduces potential biases, such as selection bias and information bias, since we relied on historical data and pre-existing samples. Additionally, without a control group of women with normal Pap and HPV DNA test results, our ability to make direct comparisons is restricted, making it difficult to definitively assess the significance of multiple HPV infections and high viral loads.

Additionally, the reliance on a retrospective sample drawn from a single center (San Martino Hospital, Genoa, Italy) with specific selection criteria may introduce selection bias, overrepresenting higher-risk cases and limiting the generalizability of findings. For example, women with normal screening results were excluded, potentially inflating the prevalence of HPV positivity and high-risk lesions in the sample. Moreover, three samples were excluded due to insufficient tissue, which could further impact generalizability.

We used a semi-quantitative method for assessing HPV viral load, which, while efficient, has certain limitations. Its semi-quantitative nature does not capture precise viral quantities, potentially overlooking subtle but clinically important variations in viral concentration. As a result, this method may be less sensitive to small changes in viral load that could influence infection persistence and progression risk. A more precise quantitative approach could provide deeper insights into the association between viral load and lesion severity. Addressing these limitations in future studies with more accurate quantitative measures would align with STROBE’s recommendation to provide detailed descriptions of measurement variables, enabling more nuanced interpretations of viral load as a prognostic factor in lesion assessment.

The absence of follow-up data regarding the persistence or recurrence of the disease is acknowledged as a limitation of the study. However, it is important to emphasize that this study was designed as a preliminary analysis aimed at evaluating the utility of HPV viral load and genotype co-infection assessments in enhancing the diagnostic accuracy of triage in cervical cancer screening. Given the retrospective nature of the study and the limited availability of follow-up information, such data could not be included. Future prospective studies are planned to incorporate follow-up assessments to better evaluate the persistence of lesions and the risk of recurrence, particularly in patients with higher viral loads or multiple HPV infections.

By recognizing these limitations, we underscore the need for further research to validate our findings and to explore additional factors that could improve the accuracy and effectiveness of cervical cancer screening protocols. Future studies should incorporate a more diverse, prospective cohort, including women with normal screening results, to provide a more comprehensive assessment of the diagnostic value of HPV viral load and co-infections. These studies should also explore the role of additional biomarkers and vaccination status in cervical cancer prevention strategies, aligning with recommendations to refine and expand screening practices. By addressing these challenges, future research can optimize risk stratification and improve outcomes for women undergoing cervical cancer screening.

## 6. Conclusions

This study demonstrates the potential of molecular markers, such as HPV viral load and co-infection status, to enhance the accuracy of triage in cervical cancer screening. Integrating these molecular markers into routine screening practices could enhance diagnostic precision, especially in differentiating true infections from non-neoplastic changes, thus reducing unnecessary invasive procedures and optimizing healthcare resources. Clinically, this approach offers a more effective stratification of patients, minimizing overtreatment of low-risk lesions and enhancing preventive screening quality.

Despite the small size of our case series, these findings suggest that assessing viral load and multiple HPV infections could be valuable biomarkers for predicting HPV persistence and lesion progression. Future research in larger, prospective cohorts is essential to validate these preliminary findings and further establish the clinical utility of viral load and genotype co-infections in cervical cancer prevention strategies.

## Figures and Tables

**Figure 1 life-15-00153-f001:**
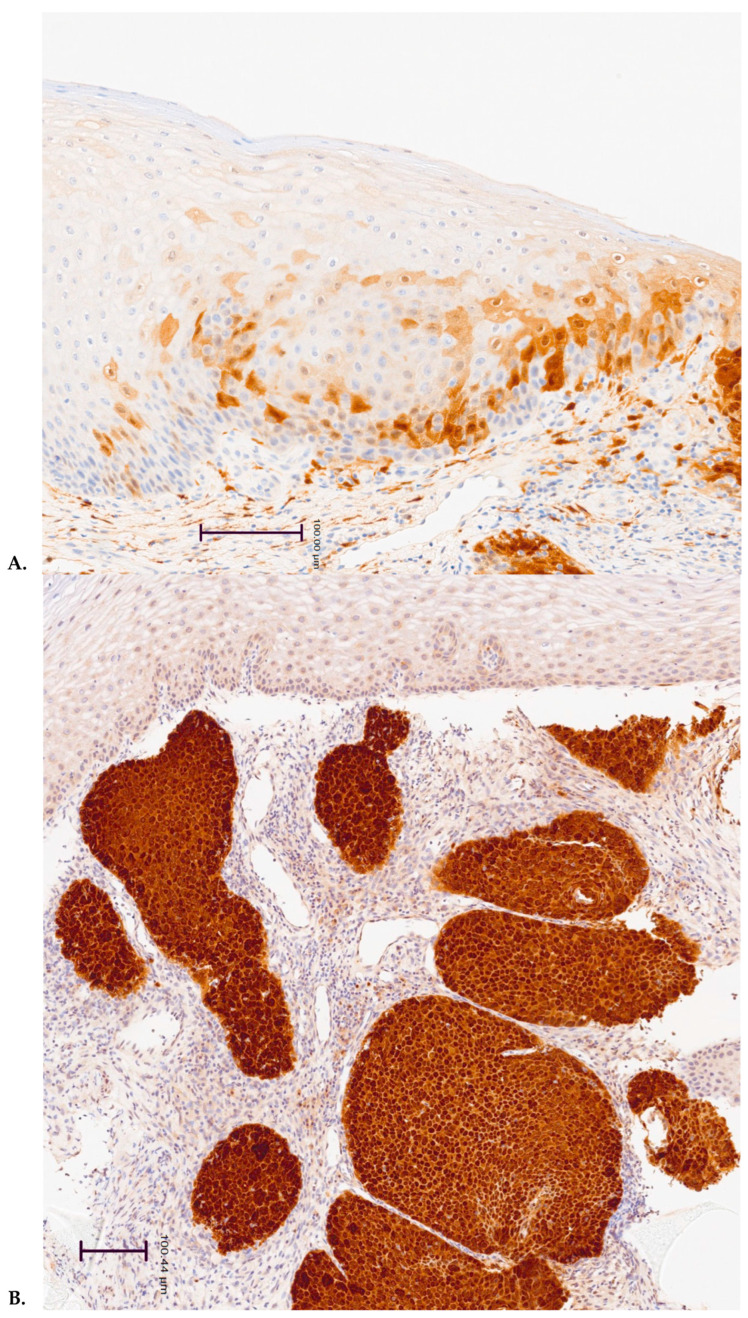
Immunohistochemical images of cervical biopsies illustrating two types of p16 positivity. (**A**) “Patchy” or focal positivity (not indicative of viral integration), showing a nest of immature metaplasia with patchy p16 staining (20×). (**B**) “Block-type” positivity (indicative of viral integration), displaying multiple foci of high-grade dysplasia with diffuse nuclear and cytoplasmic p16 staining (10×). This figure effectively demonstrates block-type p16 positivity, confirming viral integration, a critical event in HPV-related carcinogenesis.

**Figure 2 life-15-00153-f002:**
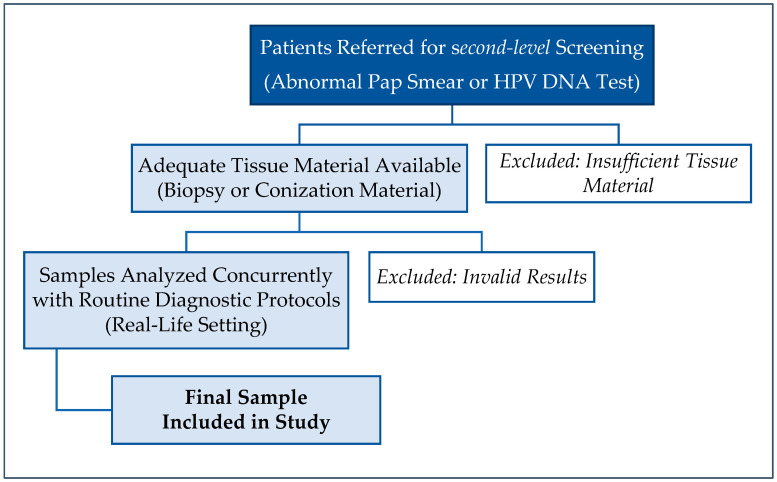
Flowchart summarizing the inclusion and exclusion criteria for this study, highlighting patient selection and analysis conditions.

**Figure 3 life-15-00153-f003:**
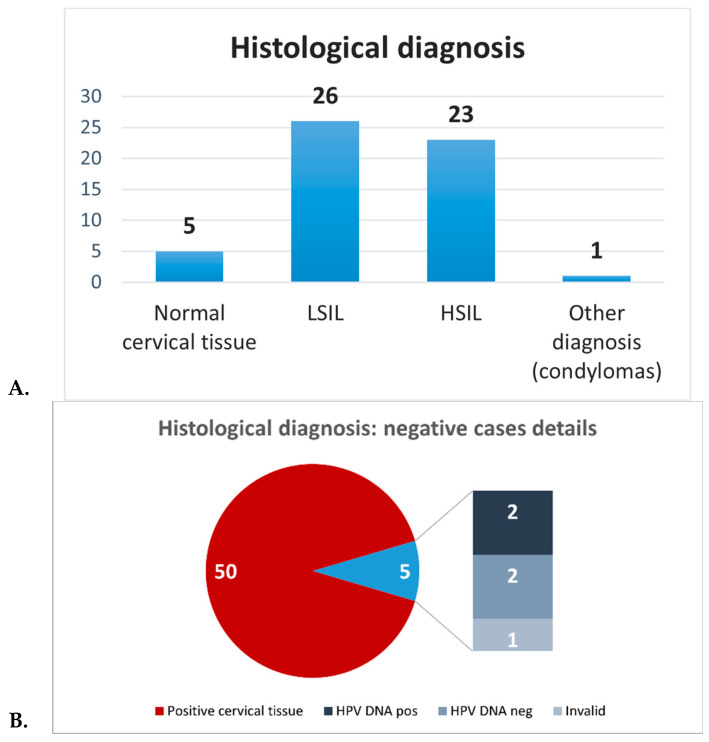
Distribution of histological diagnoses. (**A**) shows the distribution of histological diagnoses in the cervical sample cohort, showing the prevalence of LSILs and HSILs compared to normal tissue and condylomatosis. (**B**) shows the details of the negative cases.

**Figure 4 life-15-00153-f004:**
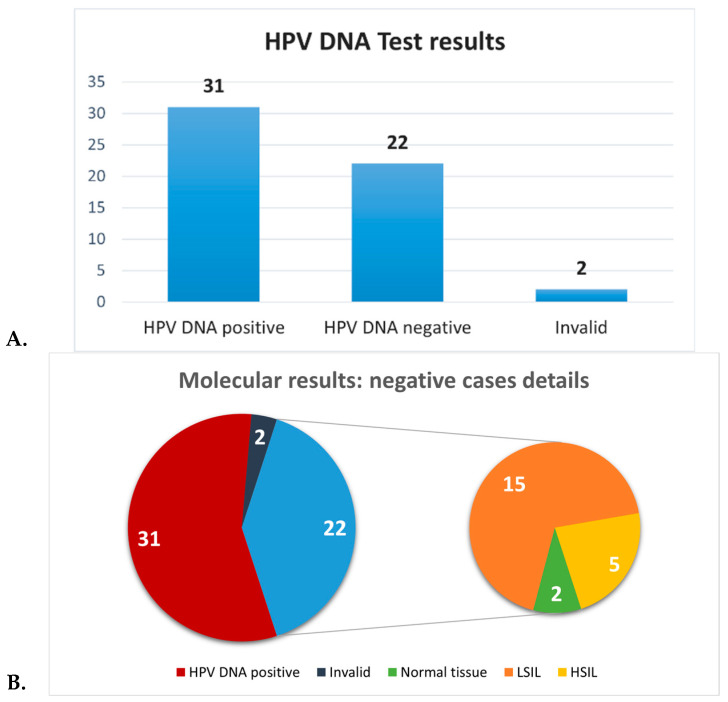
Molecular analyses. (**A**) shows the proportion of HPV DNA-positive, HPV DNA-negative, and invalid cases among the cervical samples. (**B**) shows the details of the negative cases.

**Figure 5 life-15-00153-f005:**
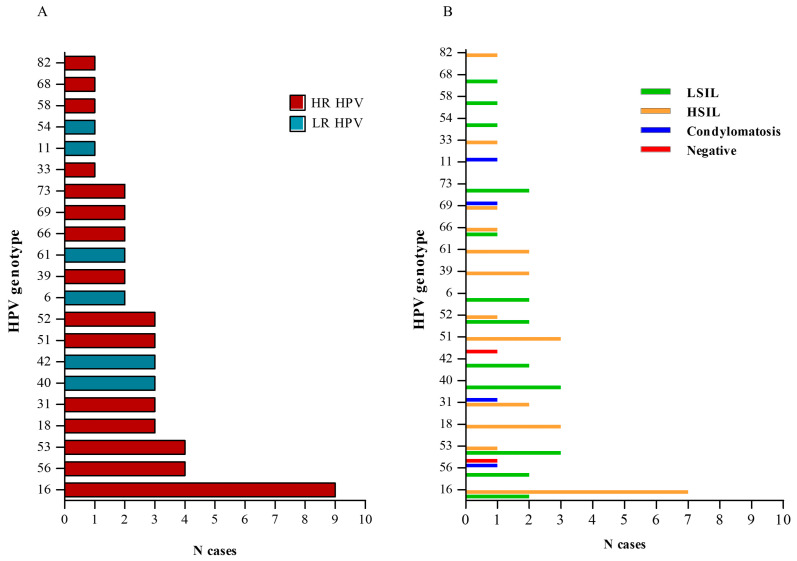
HPV genotypes and histological diagnosis. (**A**) Number of cases infected by the same genotypes; (**B**) frequency and histological diagnosis of each infecting genotype.

**Figure 6 life-15-00153-f006:**
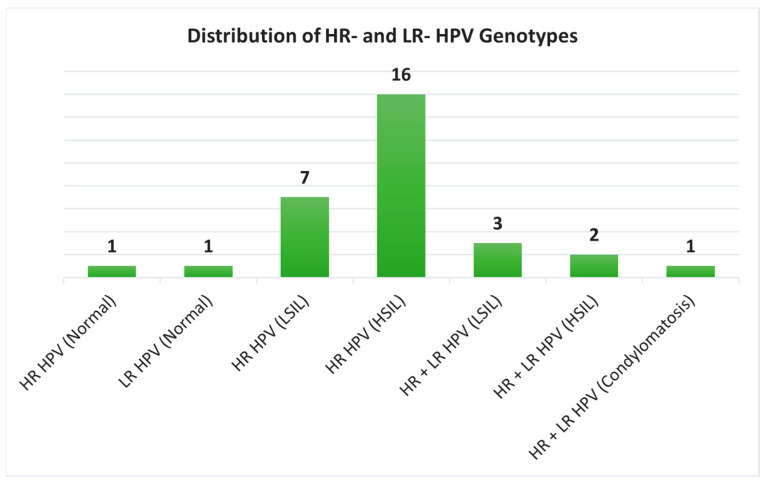
Distribution of high-risk (HR) and low-risk (LR) HPV genotypes. Distribution of high-risk and low-risk HPV genotypes across different histological diagnoses, showing a higher prevalence of HR HPV in HSILs.

**Figure 7 life-15-00153-f007:**
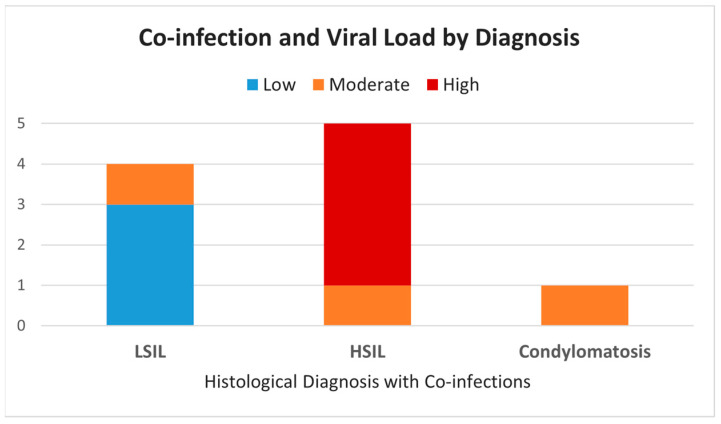
Distribution of co-infection and viral load by histological diagnosis. This figure illustrates the prevalence of multiple HPV co-infections and corresponding viral load levels across different histological diagnoses, including LSILs, HSILs, and condylomatosis. The data indicate a trend where higher viral loads (represented in red) are more frequent in higher-grade lesions (HSILs), suggesting a possible correlation between viral load intensity and lesion severity.

**Table 1 life-15-00153-t001:** Study population and sample characteristics. Table summarizing the characteristics of the study population (*n* = 55) in terms of mean age, types of samples collected, histological diagnoses, and molecular results.

Parameters	Study Population (*n* = 55)	
	*N*	% or Mean
Age (mean years)		40.5
Range	26–74	
*Sample types*		
Cervical biopsies	42	76.4
Conization	13	23.6
*Histological diagnosis*		
Normal cervical tissue	5	9.1
Abnormal cervical tissue	50	90.9
LSIL	26	47.3
HSIL	23	41.8
Other diagnosis (condylomas)	1	1.8
*Molecular results*		
HPV DNA positive	31	56.4
HPV DNA negative	22	40.0
Invalid	2	3.6

**Table 2 life-15-00153-t002:** Flow diagram for sample processing and exclusions. This flow diagram illustrates the main steps in the sampling process enhancing clarity by visualizing the sample processing steps and exclusions.

Flow Diagram		
		*N*
Sample Collection	Initial samples collected	55
Exclusions	Samples were excluded due to insufficient material	3
Valid Samples for Analysis	Samples were eligible for analysis	52
Histological and Molecular Analysis	Valid samples underwent both histological analysis and molecular analysis, including HPV genotyping and viral load assessment	52
Final Analyzed Samples	Samples completed all steps of the analysis	52

**Table 3 life-15-00153-t003:** Positivity of LR and/or HR HPV infections in the different histological samples. The table summarizes the distribution of high-risk (HR) and low-risk (LR) HPV infections across various types of histological samples. This section highlights cases positive for HR HPV only, LR HPV only, and both types (LR and HR), categorized by histological type: low-grade lesions (LSILs), high-grade lesions (HSILs), condyloma, and normal tissue.

Parameters	*N*
Total genotypes detected	21
HR HPV	15
LR HPV	6
*Histology of the samples positive for HR HPV only*	
LSIL	7
CIN 2/3	16
Condyloma	0
Normal tissue	1
*Histology of the samples positive for LR HPV only*	
LSIL	0
HSIL	0
Condyloma	0
Normal tissue	1
*Histology of the samples positive for both LR and HR HPV*	
LSIL	3
HSIL	2
Condyloma	1
Normal tissue	0

**Table 4 life-15-00153-t004:** Characteristics of co-infections, genotypes, and viral load in histological diagnoses. This table summarizes molecular analysis in cervical lesion diagnoses, detailing high- and low-risk HPV genotypes, viral load distribution, and co-infection frequencies across LSIL, HSIL, and condylomatosis cases.

Histological Diagnosis	Molecular Analysis	
	HR HPV Genotypes	LR HPV Genotypes
LSIL	52+, 56+, 73+	6++, 40++, 42++
LSIL	56+, 73+	6+++, 40++, 42++
LSIL	66+	54++, 40+++
LSIL	52+, 68++	
HSIL	31+, 39++	61+
HSIL	51+, 53++	
HSIL	31+, 39++	61+
HSIL	16++, 66++	
HSIL	18+, 51++	
Condylomatosis	31+, 56+, 69+	11+++
*Co-infections and genotype*		
	Total cases:	10
	HR HPV	13
	LR HPV	6
*Viral load and genotype*		
	*HR HPV*	
	(+)	14
	(++)	7
	(+++)	0
	*LR HPV*	
	(+)	2
	(++)	6
	(+++)	3
*Co-infection histology*		
	LSIL	4
	HSIL	5
	Condylomatosis	1

(+) weak, (++) medium, (+++) high.

## Data Availability

The datasets used and/or analyzed during the current study are available from the corresponding author on reasonable request.

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
