# Peer review of "Evaluating HPV Viral Load and Multiple Infections for Enhanced Cervical Cancer Risk-Based Assessment"

_life, 2025, doi:10.3390/life15020153_

Round 1
Reviewer 1 Report (Previous Reviewer 2)
Comments and Suggestions for Authors
No more comments
Author Response
Please see the attachment.

Reviewer 2 Report (Previous Reviewer 3)
Comments and Suggestions for Authors
I recommend publishing.
Good job.
Author Response
Please see the attachment.

Reviewer 3 Report (New Reviewer)
Comments and Suggestions for Authors
This prospective study focuses on optimization of molecular cervical screening strategies in a European setting. Italy has pioneered the introduction of HPV primary screening and has contributed vastly in the relevant literature, of added value is that this is not an industry –commissioned work, as no funding sources are disclosed. Interesting studies like this work reflect real life clinical practice in pragmatic terms and merit publication. I hope my comments are constructive and might help.
HPV viral load is an established and extensively studied biomarker with a very sound biological basis. Perhaps, the reasons for its non-widespread implementation might be traced on biotechnology industry marketing priorities. However, nowadays several HPV molecular assays have the potential to provide information on viral load with minor additional steps in the laboratory protocol or the use of upgraded software or viewers, without the need of resorting to additional molecular platforms, or recalling the patient to obtain a new sample. Furthermore, assessing both biomarkers in one integrated analyzer does not add complexity and repeatability issues. Such examples are Seegene’s Anyplex assessed in this study, or the correlation of viral load with Ct values throughout the Roche Cobas systems.
Line 3: “Cervical cancer risk assessment” is in line with the ASCCP notion of “risk-based” cervical screening and colposcopy.
Line 32: “Second level” obviously refers to secondary screening. Please do consider (as well as numerous other situations throughout the text) the term “triage” instead of “second level”. This is very important for future citations from gynecologists, epidemiologists, etc.
Line 66, The Bethesda system (TBS 2001) was updated in 2014.
Line 68, consider adding “recipients of solid organ transplants”.
Lines 83-85, until recently, each Italian prefecture could individually liberally select their own protocols for cervical screening based on current evidence - please specify.
Line 86: Does co-testing represent the usual primary cervical cancer screening strategy for most Italian prefectures currently? The literature suggests that several Italian regions/prefectures solely rely on HPV primary screening.
Lines 87-88, again, “triage” instead of “second level”.
Lines 95-96: “Regular Pap and HPV DNA testing are effective for early detection of preneoplastic lesions.”: Most global guidelines currently recommended hrHPV DNA in 3-5 year intervals.
Line 110: Re mRNA HPV, do consider adding the reference: Valasoulis G et al, Int J Molec Sci 2024, DOI: 10.3390/ijms252313146
Line 113, Re STI’s, do consider adding the reference: Valasoulis G et al, Pathogens 2023, DOI: 10.3390/pathogens12111347
Line 115; again, “triage” instead of “second level”.
Line 121, “minimize unnecessary follow-up procedures” consider “minimize colposcopy clinics overload and unnecessary follow-up procedures”
Line 128, Figure 1: An excellent, very well captured, informative and self-explainable immunohistochemical image.
Line 142, the authors might wish to explain why they opted for “selective sampling” instead of another strategy.
Line 187-198, a couple of citations quoting published literature (for instance, Martinelli, M. et al, Int. J. Mol. Sci. 2023, also originating from Italy) are needed.
Line 259 does “abnormal cervical tissue” also include specimens with chronic cervicitis, or exclusively HPV-related lesions? Please consider that chronic cervicitis without concomitant SIL is often the only identified culprit in conization pathology specimens of hrHPV DNA positive as well as from APTIMA positive women.
Line 294, “samples”, typo error, accent grave.
Line 340, instead of “it appears evident that” do consider “it was evident”.
Lines 344-345: With widespread use of extended HPV genotyping, more published data on the role of multiple HPV genotypes on CIN development are anticipated.
The Discussion section, albeit extensive and with interesting remarks, largely lacks in coherence, as if several authors contributed with a couple of paragraphs. A more uniform style avoiding repetitions could be adopted. Some benchmark studies from the literature regarding the correlation of lesion severity with a) multiple HPV genotypes as well as b) viral load would be welcome in this context.
Line 347, again instead of “second line” please consider “triage”.
Lines 379-380 are repeated in lines 395-396.
Line 399, a reference should be quoted after “pathologists”.
Line 422: a reference suggestion (probably) by a previous reviewer is incorrect and needs formatting.
Line 423: Prof. Joakim Dillner also recently commented on the low HPV18 prevalence (PMID: 39105357) most likely attributed to HPV vaccination herd affects.
Line 434: Does all the text from line 457 onwards belong to the “Study Limitations” section? If so, this section (Study Limitations) is disproportionally lengthy, has several repetitions and in essence most of its content could be relocated in the main Discussion section. The paragraph from line 483 deals with the strengths of the study, however there is not such title.
Lines 473-474: Roche’s Cobas systems (which however do not offer extended HPV genotyping) correlate viral load & cycle threshold (Ct) values (Duan et al 2020, Song et al 2020, Inturrisi et al 2021, Arbyn et al 2023, Lei et al 2024). How does Seegene’s Anyplex platform studied here relatively compare?
References 19-22 have a different format; they have been probably suggested or added by previous reviewers.
Author Response
Please see the attachment.

This manuscript is a resubmission of an earlier submission. The following is a list of the peer review reports and author responses from that submission.
Round 1
Reviewer 1 Report
Comments and Suggestions for Authors
This retrospective study evaluated the utility of real life assessment of viral load in cervical cancer screening. The inclusion of patients is not clear. It is not clear why a period between January and June 2021 was selected? In addition, what were the exclusion criteria? The patients' selection should be presented in a flowchart. Finally, the number of included patients is too small to draw any definite conclusions
Reviewer 2 Report
Comments and Suggestions for Authors
Comments:
1. Please discuss the main issue of small sample size.
2. Figure 1: please add scale bar on images.
3. Table 1: Please add smoking and alcohol history.
4. Any data with p53 in this study?
5. Any patients have other cancers in this study?
6. Any data of CDK levels in this study?
Reviewer 3 Report
Comments and Suggestions for Authors
I congratulate the author of the work which addresses the issue of multiple HPV infections in the development of SIL lesions. In this study, a rarely used method for detecting HPV infections from paraffin blocks was used.
The study group is not large but sufficient for publication.
I have a few comments on the work.
Keywords: add: HPV from formalin-fixed paraffin blocks. FFPE
The authors should use the LATS terminology in combination with the CIN terminology, which is very important and transparent: LSIL – CIN 1, HSIL – CIN 2, CIN 3?
In the introduction, please write about the limitations of the method of obtaining material for PCR testing from paraffin blocks.
In the introduction, please mention that there are already other diagnostic methods such as mRNA, E4, DS or methylation tests. See below:
Przybylski et all Expression of E4 Protein and HPV Major Capsid Protein (L1) as A Novel Combination in Squamous Intraepithelial Lesions.
Rokita et all Comparison of the effectiveness of cytodiagnostics, molecular identification of HPV HR and CINtecPLUS™ test to identify LG SIL and HG SIL
Research confirms that HPV type 18 is becoming less common.
Currently, HPV type 18 is no longer so common. Please see the publication like: Clinical use of the Onclarity test with extended HPV genotyping and phenotyping in patients with suspected squamous intraepithelial lesions. DOI:10.5603/gpl.96712 Ginekol Pol 2024;95(5):328 -334.
There is no information whether the patients were vaccinated or not - and this is very important. If vaccinated with what vaccine. Vaccination affects the recurrence of the disease and persistent infections.
Figure 1 is very nice, but is it related to the topic of the work?
The ISH technique can obtain HPV staining in tissues. See: https://www.mdpi.com/2072-6694/16/20/3485
Very nicely presented tables and grafts - good point!
I recommend publishing it after making corrections.
Good job.
Reviewer 4 Report
Comments and Suggestions for Authors
Dear Author, thanks for the opportunity to review this manuscript.
Abstract: please consider revision of abstract in a structured form (Aim, Methods, Results, and Conclusions).
Introduction: Please consider the length reduction of this section because, in my opinion, some sentences are redundant (for example rows 51-59).
Material and methods: What are the criteria to establish an indication for biopsy or conization? Why Did you choose six months?
Results: In my opinion, the sample should be enlarged to improve the quality of the findings.
"Molecular analyses revealed positivity for HPV DNA in 31 cervical samples (56,4%) whereas 22 specimens were HPV-negative (40%)". Why these patients underwent biopsy or conization if HPV was negative?
Discussion:
"The most relevant finding of our work was the fact that almost half (40%) of the cervical samples collected from women performing a second-level cervical screening program, were negative at the molecular analysis".
Why Can you explain this? Is it a bias of selection or a pre-treatment low quality HPV test?
There are not follow-up data concerning the persistence or recurrence of the disease. Please, you have to report follow-up data, especially in the patients with higher viral load.
Suggested citation: Mereu, L.; Pecorino, B.; Ferrara, M.; Tomaselli, V.; Scibilia, G.; Scollo, P. Neoadjuvant Chemotherapy plus Radical Surgery in Locally Advanced Cervical Cancer: Retrospective Single-Center Study. Cancers2023,15,5207.
Comments on the Quality of English Language
Minor English revision should be performed.